# Development and Validation of Nutrition and Food Safety Educational Material for Fish Processors in Nigeria

**DOI:** 10.3390/ijerph20064891

**Published:** 2023-03-10

**Authors:** Grace Adeola Adegoye, Terezie Tolar-Peterson, Henrietta Nkechi Ene-Obong, Joseph Nkem Nuntah, Monica M. Pasqualino, Rahel Mathews, Juan L. Silva, Wen-Hsing Cheng, Marion Willard Evans, Lauren Pincus

**Affiliations:** 1Department of Nutrition and Health Science, Ball State University, Muncie, IN 47306, USA; grace.adegoye@bsu.edu; 2Department of Food Science, Nutrition and Health Promotion, Mississippi State University, Starkville, MS 39762, USA; rm933@msstate.edu (R.M.); jsilva@foodscience.msstate.edu (J.L.S.); wcheng@fsnhp.msstate.edu (W.-H.C.); 3Department of Health Science and Human Ecology, California State University San Bernardino, San Bernardino, CA 92407, USA; 4Department of Human Nutrition and Dietetics, University of Calabar, Calabar 540271, Nigeria; henriettaeneobong@unical.edu.ng; 5Department of Aquaculture and Fisheries Management, University of Benin, Benin 300213, Nigeria; josy_joeng@yahoo.com; 6WorldFish One CGIAR, Jalan Batu Maung, Bayan Lepas 11960, Malaysia; m.pasqualino@worldfishcenter.org (M.M.P.); lpincus@gmail.com (L.P.); 7Center for Human Nutrition, Department of International Health, Johns Hopkins Bloomberg School of Public Health, Baltimore, MD 21205, USA; 8College of Nursing and Health Professions, The University of Southern Mississippi, Hattiesburg, MS 39406, USA; willard.evans@usm.edu

**Keywords:** content validity, educational material, nutrition, food safety, literacy, low- and middle-income countries

## Abstract

Introduction: fish can be an affordable and accessible animal-source food in many Low- and Middle-Income Countries (LMIC). Background: Traditional fish processing methods pose a risk of exposing fish to various contaminants that may reduce their nutritional benefit. In addition, a lack of literacy may increase women fish processors’ vulnerability to malnutrition and foodborne diseases. Objective: The overall aim of the project was to educate women and youth fish processors in Delta State, Nigeria about the benefit of fish in the human diet and to develop low literacy tools to help them better market their products. The objective of this study was to describe the development and validation of a low-literacy flipbook designed to teach women fish processors about nutrition and food safety. Method: developing and validating instructional material requires understanding the population, high-quality and relevant graphics, and the involvement of relevant experts to conduct the content validation using the Content Validity Index (CVI) and the index value translated with the Modified Kappa Index (*k*). Result: The Item-level Content Validity Index (I-CVI) value of all domains evaluated at the initial stage was 0.83 and the Scale-level Content Validity Index (S-CVI) was 0.90. At the final stage, the material was validated with CVI 0.983 by four experts and satisfied the expected minimum CVI value for this study (CVI ≥ 0.83, *p*-value = 0.05). The overall evaluation of the newly developed and validated flipbook was “excellent”. Conclusions: the developed material was found to be appropriate for training fish processors in Nigeria in nutrition and food safety and could be modified for a population of fish processors in other LMICs.

## 1. Introduction

Nutrition literacy is the degree to which individuals can obtain, process, and understand nutrition information and skills needed to make appropriate nutrition decisions [1,2]. It is a strategy for improving the quality of nutrition, diets, and food security [3,4]. Food security is defined as “when people at all times have physical, social, and economic access to sufficient, safe, and nutritious food to meet dietary needs and food preferences for an active and healthy life” [5]. Food safety is a global public health concern [6]. Awareness of nutrition and food safety is crucial in disease prevention [7].

Nutrition and food safety literacy is of global importance and could be considered an integral component of food security. Unfortunately, it is not fully appreciated by many public health authorities despite a global increase in the prevalence of foodborne illnesses [8]. Food literacy is “understanding nutrition information and acting on it in a way that is consistent with nutrition goals and food wellbeing” [9]. Therefore, teaching nutrition and food safety may be an effective way to improve the nutritional value, quality, and safety of food products. A recent study documented that food safety instruction creates a positive shift in the knowledge paradigm, stimulates a behavioral change towards safe food handling, and minimizes the risk of foodborne illnesses [10]. Health education and promotional tools are effective interventions for modifying health behavior [11]. Printed educational materials such as modules and flipbooks enhance learning, facilitate the delivery of key messages in an entrancing mode, serve as reminders and reinforcement for oral communication [12], and improve knowledge, satisfaction, and adherence to health instruction [13]. 

The educational level or literacy status of the audience or target population is important to consider when providing nutrition and food safety instructions. Despite the relatively high educational attainment in Nigeria, with 45% of women and 62% of men having completed secondary education or higher [14], malnutrition and foodborne diseases remain prevalent and contribute to increased food insecurity. Therefore, there is a need for nutrition and food safety literacy education. 

Nigeria’s tertiary institution’s curriculum for the training of fishery graduates covers fishery technology, processing and storage, and fish nutrition [15]. However, to the best knowledge of the authors, there is no validated, easy-to-understand nutrition and food safety material for training small-scale fish processors in order to improve the quality and safety of processed fish products in Nigeria. 

This study was developed as part of the Feed the Future Innovation Lab for Fish (FIL), the US Government’s hunger and food security initiative funded by the United States Agency for International Development (USAID). This initiative aims to reduce poverty and improve nutrition, food security, and livelihoods in low-income countries by supporting projects in sustainable aquaculture and fisheries [16], as well as working towards achieving the Sustainable Development Goals 1, 2, and 5 [17] through innovation central to advancing novel solutions that support goals that reduce global hunger, poverty, and undernutrition and promote quality education and gender equality. 

Malnutrition, food insecurity, and poverty remain persistent public health concerns in sub-Saharan African countries [18]. Aquaculture and fisheries have been identified as indigenous resources with the potential to provide a solution to these problems because they serve as a wellspring of animal-based proteins and a source of income for many people [19,20], including marginalized women and youths that are culturally deprived of employment, higher pay, and education. Despite the potential capacity of the aquaculture and fisheries system to improve nutrition, food security, and livelihood, most African countries remain impoverished and malnourished [21] as a result of poor handling, processing, storage, and preservation. Training is an identified need [22] to address the lack of formal education on safe fish handling and processing among small-scale or artisanal fish processors in low-income countries. A study conducted in Northern Nigeria showed that women employed in small-scale fishing and processing businesses have limited access to extension services and training, capital, and modern technology [23]. Little or no education is a factor that affects the acquisition and adoption of improved fish processing techniques [24]. Improved food quality could prevent malnutrition and micronutrient deficiencies that often result from low-quality diets. The food safety knowledge and handling practices among the stakeholders influence the quality and safety of fish products available for the consumers [25]. Contaminated food products (a result of poor handling and processing) can lead to food-borne disease outbreaks. A study in Nigeria showed that a majority of fish vendors lacked knowledge about microbial contamination and the risks associated with unhygienic fish handling [26]. 

Fish processing methods in low-income countries remain limited to traditional salting, sun-drying, and smoking [27,28]. These methods provide some level of preservation, require no technical know-how or skills, and are cost-effective. However, these methods are not sustainable because of possible exposure to contaminants such as smoke, Polycyclic Aromatic Hydrocarbon (PAH) [29,30], dust, sand, insects, dirt, pests, microorganisms (bacterial, mold), and others during and after processing [31,32]. This contributes to increased post-harvest waste with a consequential effect on food security, nutrition, and the economy.

The USAID Feed the Future Innovation Lab for Fish (FTF FIL) aimed to enhance nutrition and food security among sub-Saharan African countries including Ghana, Uganda, Kenya, Zambia, Nigeria, and Malawi [16,33]. This study is one of the innovative strategies of the Fish Innovation Lab (FIL). It addresses the objective of a project implemented in Delta State, Nigeria (Nourishing Nations: Improving the Quality and Safety of Processed Fish Products in Nigeria) to both educate women and youth fish processors in the Delta State about the benefit of fish in the human diet and to develop low-literacy tools which can help better market their products. This objective was accomplished by developing training materials to improve knowledge in nutrition, food safety, and fish processing methods as a sustainable contribution to improve the quality of processed fish, nutrition, livelihood, and income.

The Nourishing Nations project was implemented in collaboration with the International Center for Living Aquatic Resources Management (ICLARM), also known as WorldFish. In addition, the Food and Agricultural Organization of the United Nations, in partnership with academia and research institutions (Mississippi State University FTF), also aimed at the SDGs 2, 12, and 14 [34] to improve food and nutrition security and alleviate poverty among the vulnerable population through sustainable aquaculture development efforts. 

Training, workshops, and sustainable projects are part of the innovative approach sponsored by the USAID FTF FIL to achieve its goal, which requires designing and developing training material appropriate for the target population.

Developing suitable, comprehensible, culturally appropriate, and relevant training material is important for improving nutrition and food safety knowledge, food handling behavior, and quality fish production. This study aimed to develop and validate low-literacy materials to teach fish processors in Delta State, Nigeria about nutrition and safe fish handling and processing to address knowledge gaps, food insecurity, poor quality processed fish products, and food safety issues. To the authors’ knowledge, no validated nutrition and food safety low-literacy material is presently available in Nigeria for training fish processors in these areas. The developed and validated nutrition and food safety flipbook will be a useful tool in the interactive training of small-scale fish processors.

## 2. Methods

### 2.1. Curriculum Development

The first step in the process of developing low-literacy educational materials was the curriculum development (Figure 1). Curriculum themes were designed to meet the objectives of the Nourishing Nations project which focused on improving nutrition knowledge and the quality of processed fish products in Delta State, Nigeria. Designed curriculum included collaborative or complementary topics on nutrition and food safety to form content focused on improving nutrition knowledge, safe fish handling, and processing among fish processors. Table 1 presents the content of the nutrition and food safety flipbook, which contained the following modules: (1) healthy eating; (2) animal-source food, with a focus on fish for human nutrition; (3) food safety; (4) fish processing techniques; (5) food poisoning and contamination, with a focus on fish contamination; (6) hygiene practices; (7) the economic and nutritional benefits of fish consumption. Specific objectives were formulated for each module to maximize achievement toward the set goal of improving nutrition and food safety awareness among women fish processors.

### 2.2. Review of the Relevant Literature

Selecting relevant scientific information for content development involved a literature review of books, periodicals, and publications on nutrition, food safety, safe fish handling, processing, hygiene, and sanitation. The low-literacy training material was developed and prepared in English and written at the 4th-grade reading level. It contained few words with adequate information and appropriate illustrations, pictures, and high-quality and culturally appropriate infographics [35]. Jargon and technical terms were avoided to achieve high readability and comprehensibility [36]. A draft of the material was created using Microsoft Word and PowerPoint. A high-resolution camera and smartphone were used to capture graphics. The seven-module flipbook was developed with an average of ten slides per module and titled: “Nutrition education, food safety, and safe fish handling practice guide for fish processors in Nigeria” (Appendix A). 

### 2.3. Selection of Experts 

A content evaluation panel is a group of experts that individually validate each item of the material at the initial stage and collectively evaluate at the final stage [37]. The selection of experts was guided by recommendations for the minimum acceptable CVI values (Table 2). The selected group of experts included nutritionists or dietitians, public/environmental health specialists, low-literacy education experts, fisheries and fish value chain experts, and food safety experts. Selecting members from different geographical locations can raise the chance of recognizing colloquial terms inappropriate for an instrument [38]. Twelve experts were therefore selected representing Nigerian and USA nationalities using well-defined criteria including areas of expertise, experience, and qualifications [39]. An introduction letter was sent to each expert to solicit the panel’s participation. Four of the invited panelists declined participation based on either conflict of time or interest. Eight accepted, but only six completed the assignment at the initial stage of the content validation. Four were invited for the final content validation. The number of panelists that participated in this study aligned with the expert recommendation [39,40].

### 2.4. Content Validity Index 

After panel members accepted the task, they were sent the educational material in a PowerPoint format with the accompanying multiple-choice questions for each module. Each panelist was provided with one Content Validation Index (CVI) assessment form via email that contained ten domains for each of the seven modules of the low-literacy flipbook. The ten domains, identified and modified based on the Taveres model [40] to suit this study, included: (1) objective, (2) content, (3) relevance, (4) language, (5) infographics, (6) design, (7) motivation, (8) culture, (9) methodology, and (10) pre-and post-quiz test. Panel members were asked to indicate their agreement through a five-point Likert scale to determine the relevancy of each of the ten domains. 

Panelists outside of the United States were requested to send their CVI report forms through email, while those within Mississippi State University submitted their report forms in person. Panelist judgments were analyzed by computing the Item-level Content Validity Index (I-CVIs) and the Scale-level Content Validity Index (S-CVIs) to determine the relevancy of the items in the domains. I-CVI was also compared with the Modified Kappa Index (*k**); this is an index of agreement among the panelists that the item is relevant and is categorized as fair, good, or excellent [41]. 

CVI was used to determine the degree of usefulness of each component of the training material. Using a content validity panel of six members, a minimum value of 0.99 was required for the CVI at *p*-value = 0.05 [42]. In this study, the expected minimum CVI value based on the number of experts involved was between 0.83 and 1.0. The higher the percentage of the panelists’ agreement on the relevancy or essentiality of the evaluated item, the greater the degree of its content validity. The Content Validity Ratio (CVR) was used in determining the rejection or retention of specific items.

The content validation data collected were entered in Microsoft Office Excel (Microsoft Corporation) and analyzed using the CVI [41]. 

The study was determined exempted by the Mississippi State University Institutional Research Board (IRB).

## 3. Results 

### 3.1. Content Validity Index 

Educational materials and instructions written with advanced English or above fourth-grade reading level may be too technical and difficult to understand by a low-literate population and this may likely jeopardize the validity and credibility of such instruments. A study from the Bangladesh perspective shows that comprehensible English learning material improves students’ performance [44]. Another study in Morocco reveals an advantageous impact of comprehensive inputs on English learners [45]. 

The result of this study shows that the developed flipbook has a high Content Validity Index, which is an indicator of the appropriateness, comprehensibility, and adaptability of the material by the low-literate population. 

Appendix A presents the results of the initial and final validation by six and four panelists, respectively. The I-CVI value of all domains evaluated at the initial stage was 0.83, except for the infographic domain in module one, which was 0.81, and the culture domain in module two, with a value of 0.77. The S-CVI for the initial validation was 0.90 and increased to 0.983 at the final validation after making necessary adjustments based on the panelists’ recommendations, as summarized in Table 3. The recommendations helped to improve the cultural appropriateness of the newly developed material from the I-CVI value of 0.77 to 0.92. 

### 3.2. I-CVI and Modified Kappa Index translation

Table 4 presents the I-CVI evaluation table and the number of panelists agreeing using six and four experts. It also shows the computed probability of chance occurrence (*P_c_*) based on the number of panelists (N) and the number agreeing on relevance (A) to determine the kappa designating agreement on relevance (*k**) and compared with the evaluation criteria for kappa (E_K_). 

Overall, I-CVI = 0.67 when four out of five or four out of six of the panelists rated an item as relevant; I-CVI = 0.83 when five out of six rated an item as relevant; I-CVI = 0.75 when three out of four rated an item as relevant; I-CVI = 1 when all the panelists rated an item as four.

The minimum I-CVI in the final validation using four experts was 0.75, *k** value = 0.67, and E_k_ evaluation description as “good”. The maximum content validity value that could be achieved = 1, *k** value −1.00, and E_k_ evaluation description as “excellent”. 

## 4. Discussion

Content validity is the degree of agreement or intersect between the performance of the material under evaluation and the ability to function in the job performance domain. The relevancy of each component of the flipbook was determined using the Content Validity Index (CVI). 

Based on the results of this study, the developed material was found to be appropriate for training fish processors in Nigeria in nutrition and food safety. The Item-level Content Validity Index (I-CVI) value of all domains evaluated at the initial stage was 0.83 and the Scale-level Content Validity Index (S-CVI) was 0.90. At the final stage, the material was validated with CVI 0.983 by four experts and satisfied the expected minimum CVI value for this study (CVI ≥ 0.83, *p*-value = 0.05). The overall evaluation of the newly developed and validated flipbook was “excellent”. 

The development and validation of a new training material is a process that includes curriculum development, objective formulation, a review of the literature, content development [35,36], and judgment quantification. 

The educational material was modified based on the CVI value. The panelists’ recommendations were instrumental in improving the overall content, language, and cultural appropriateness of the nutrition and food safety flipbook. 

The CVI value for the nutrition and food safety flipbook also satisfied Davis’s recommendation [46] that a new content validity instrument should have a minimum S-CVI of 0.80. In addition, Polit and Beck [47] recommended that an overall scale could be judged as having excellent content validity if it would be composed of items with I-CVIs that meet Lynn’s criteria [43,48] (I-CVI = 1.00 with 3 to 5 panelists and a minimum of I-CVI of 0.78 for 6 to 10 panelists and an S-CVI/Ave of 0.90 or higher). 

The overall evaluation description of the new flipbook was evaluated as “excellent” [41]. In this study, we compared the *Pc* = 0.031 (Table 4) with the *Pc* value 0.041 [41]. Although both values were still within the kappa range for excellence, the I-CVI comparison with the Modified Kappa Index and the evaluation criteria described guideline by Cicchetti and Sparrow [41] shows that the content validity using six and four experts was “good” when *k* is between 0.60–0.74, and “excellent” when *k* is >0.74.

This study is the first to develop and validate a flipbook that imparts basic nutrition and food safety information to small-scale fish processors in low-income countries. The Content Validity Index of other educational materials was considered for evaluation and compared with the validity index of the nutrition and food safety flipbook. Results of a recent study that describe the validity of android-based learning media developed using the Aiken formula for content validation, show that an index value > 0.750 is considered high validity [49]. Another study, aiming to develop and describe the quality and practicality of the measurement instrument for college students’ entrepreneurial skills, showed that content validity > 0.700 is valid based on the experts’ judgment using the Aiken V formula [50]. A closely related educational booklet for healthy eating for pregnant women had a Content Validity Index of 0.91 [13]. In this study, the CVI of the educational flipbook on nutrition and food safety for small-scale fish processors has a Content Validity Index of 0.983 and is translated, with a Modified Kappa Index (*k*) > 0.74, as “excellent”.

The validated material was tested in training 122 women and youth fish processors in three senatorial districts in Delta State, Nigeria in a 3-day interactive training using a train-the-trainer approach [51,52]. Knowledge evaluation of the participants using pre- and post-assessment quizzes showed a positive shift in knowledge paradigm among the fish processors [53].

### Strengths and Limitations of the Study

The strength of the study was the variety of disciplines of the panelists, whose areas of expertise and recommendations were found essential in the development of the material. In addition, the heterogeneity of cultural backgrounds reduces cultural biases and improves the cultural appropriateness of the newly developed nutrition and food safety flipbook. The number of panelists at the initial and final validation stage was within the recommended value to achieve the minimum acceptable CVI values for a newly developed material. The encompassing quality of the flipbook has the potential to provide the user with diverse integrated knowledge on eating healthy, benefits of fish consumption for human nutrition, safe food handling, improved fish processing methods, prevention of food contamination, hygiene practices, and economic potential of quality fish consumption presented in the seven modules of the flipbook. 

A limitation in the development of the flipbook was the westernized graphics and pictures when culturally familiar photo illustrations were not available. In the future, the researchers will engage the expertise of professional artists and graphic designers in drawing appropriate and suitable images or pictures that depict the cultural values of the targeted population. 

## 5. Conclusions

Creating educational material involves presenting key points in an easy-to-read format, high-quality graphic aids, and contributions of experts from relevant professions. The newly developed low-literacy seven-module flipbook on nutrition and safe fish handling and processing for fish processors in Nigeria was successfully validated and considered suitable and culturally appropriate for the target population. The flipbook has the potential to contribute to the improvement of knowledge about nutrition, healthy eating, dietary diversity, food security, and animal-source food in an effort to mitigate malnutrition among children, young female adolescents, and women in low- and middle-income countries. The educational material would also help to improve long-practiced fish processing methods, food safety, and food handling practices among women fish processors in Delta State, Nigeria. Finally, the newly developed and validated flipbook will be available to the public in a printable and downloadable form for teaching low-literacy fish processors nutrition, safe fish handling, and processing.

## Figures and Tables

**Figure 1 ijerph-20-04891-f001:**
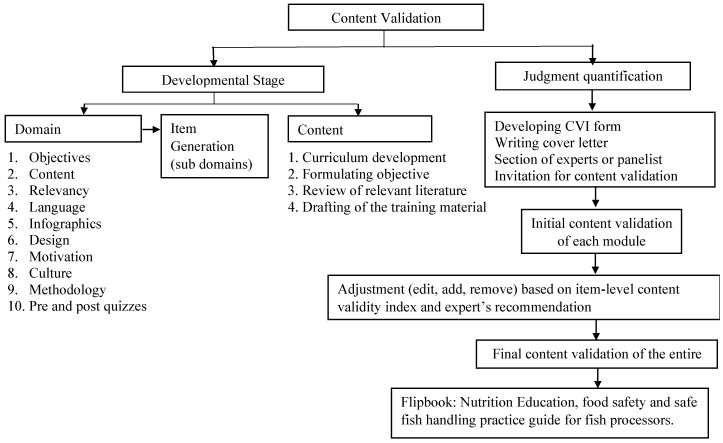
Flowchart for the content validation process. (This illustration is the detailed process involved in the development of the nutrition education, food safety, and safe fish-handling practice guide for fish processors. The process design was the authors’ perception based on a review of the existing literature on content validity and development commonly used in the nursing and health field).

**Table 1 ijerph-20-04891-t001:** Nutrition and Food Safety Curriculum and Content.

Modules/Topics	Lesson Outline (Content)	Objectives	Key Learning Areas
Module 1 Nutrition education *healthy eating habits*	What is healthy eating? Healthy diet. Benefits of eating healthy. Benefits of breastfeeding. Choose MyPlate for Nigeria: fruits, vegetables, proteins, grains, dairy, roots, and tubers. Dietary diversity: how to make a healthy meal. Summary and evaluation pre and post quiz.	(i) Understand the importance of eating healthy. (ii) Identify better food choices and combinations.	Healthy eating, eating a variety of foods to optimize nutrition.
Module 2 Animal source protein. *Fish nutrition*	Identify animal-source proteins. Nutritional value of fish. Health benefits of fish consumption for: infants and children, pregnant and breastfeeding women, adults. Dietary recommendations for eating fish. Summary and evaluation pre and post quiz.	(i) Understand the benefits of eating fish. (ii) Learn about variety of foods that are good for growth and health.	The potential of fish in reducing micronutrient deficiencies among children and women of reproductive age.
Module 3 Food safety: *fish safety and handling*	Define food safety. Keys to food safety. Foodborne illnesses. Safe fish-handling practices. Unsafe conditions that spoil fish. Safe practices: handwashing, personal hygiene, personal protective wears. Summary and evaluation pre and post quiz.	(i) Understand the concept of food safety. (ii) Understand the consequences of unsafe food handling.	Why is food safety important? Introduce food safety focusing on safe fish handling.
Module 4 Fish processing: *fish processing techniques*	Fish processing methods: traditional methods. Modern methods: local and new processed fish products. Fish processing: preparation and procedure. Summary and evaluation pre and post quiz.	(i) Learn a better and safer method of fish processing. (ii) Recognize the advantage of new methods in improving the quality of fish products.	Safe and quality fish processing techniques. Introduce food processing and focus on improved (safe) fish processing techniques and their impact on quality, safety, and nutrition.
Module 5 Food poisoning: *fish poisoning and contamination*	Define food poisoning. Identify fish contaminants. Classification of contaminants: biological, chemical, and physical. Sources of fish contamination: 10 Fs concept: flies, fingers, fork, fomites, fluid, foe (pests), fumes, field, feces, and Fahrenheit (temp). Safety guidelines for pesticide use. Health implications of fish poisoning and contamination. Preventive measures. Summary and evaluation pre and post quiz.	(i) Identity fish contaminants and health risks. (ii) Know the preventive measures.	Preventive measures. Introduce food poisoning and focus on how to prevent or avoid food poisoning contamination.
Module 6 Hygiene rules and good practices: *hygiene rules for fish handlers*	Hygiene rules. Sanitary requirements for fish processing. Personal hygiene, sanitation, and disinfection. Good practices: good hygienic practices, aquacultural, harvest, transport, processing, handling, packaging, and storage practices. Summary and evaluation pre and post quiz.	(i) Know the importance of hygiene and sanitation. (ii) Apply good practices in fish processing.	Good practices: emphasis on personal and improved food hygiene practices of fish processors. Introduce food safety rules and focus on safe fish handling, food hygiene regulations, and practices.
Module 7 Economic benefits of *quality and safe fish products*.	Fish quality. Fish loss and waste in the value chain. Poverty reduction. Economic empowerment. Improved nutrition and dietary diversity. Improved health and wellbeing. Summary and evaluation pre and post quiz.	Understand the economic benefits of quality and safe fish products for an individual and family.	Economic empowerment through quality fish production. Introduce economic benefits of quality, nutritious, and safe fish products.

**Table 2 ijerph-20-04891-t002:** The number of experts and acceptable cut-off CVI score.

Number of Experts	Min. Acceptable CVI Values	Source
3–5	1.00	[41,42,43]
6–8	≥0.83	[41,43]
9	0.78	[42,43]

The number of panelists and the corresponding degree of agreement acceptable for the cut-off CVI score.

**Table 3 ijerph-20-04891-t003:** Summary of the qualitative analysis of the experts’ recommendations.

	Recommendations of the Expert
Module 1	Increase the text font size and sizes of the pictures. Use appropriate colors. Replace dairy with milk and use meals or plates instead of diet. Replace milk in the suggested MyPlate for Nigeria with another source such as soy products or available substitutes.
Module 2	Include a picture of a well-nourished mother with a healthy child. Use a clear image to show the benefits of the fish. Use appropriate child images and words, change child to infants or baby. Move the “Benefits of breastfeeding to infants and mothers” to Module 1.
Module 3	Use more visible, culturally appropriate, and relatable pictures. Quiz #2: What are safe practices? Change TV series to Watching TV. Quiz #3: Option A is too long, keep the answers or options brief and precise.
Module 4	Number the items on the slides rather than bullets for easier reference. On slide 5, remove the statement “excess salt intake may increase the risk of high blood pressure” because it is not relevant to the module. Reorder slides on fish processing and procedures (15–17). Quiz #2: Keep options brief and concise. Do not trick the participants.
Module 5	Increase the eligibility on slide 1, increase the spacing and the font size. Label the pictures on slides 4–7. This will enhance learning. Create separate slides for the biological contaminants and biological carriers of diseases. Replace iodine with antiseptics with “open wounds on your hands” and consider using forks and a spoon when handling fish.
Module 6	Generally, font size should be increased. Separate sanitary requirements of fish processing premises from health requirements for fish processors. Check the dilution formula and change the chlorine to water volume. Quiz #1: remove the word “except” from the question, provide one correct option, and do not try to trick your audience with low literacy.
Module 7	Emphasize the economic benefit of a quality fish product. Use a brighter color to enhance the readability of the content. Slide 8 content is more relevant to food safety. Reconstruct Quiz 1 to health benefits of quality and safe fish products. Change Quiz 2 to economic benefits of quality and safe fish products. Quiz # 3: You can save money by reducing the fish waste generated: (a) yes, (b) maybe, (c) I do not think so. The options are relative and subjective. Use options yes, no, and I don’t know instead.
**Cover**	Use culturally appropriate images to enhance acceptability and inclusiveness.

**Table 4 ijerph-20-04891-t004:** I-CVI evaluation table and number of experts in agreement.

Number of Experts	The Number Giving 4 or 5 Rating	I-CVI	*P_c_*	*k**	E_K_
3	3	1.00	0.125	1.00	Excellent
3	2	0.67	0.375	0.47	Fair
4	4	1.00	0.063	1.00	Excellent
4	3	0.75	0.25	0.67	Good
5	5	1.00	0.031	1.00	Excellent
5	4	0.80	0.156	0.76	Excellent
6	6	1.00	0.016	1.00	Excellent
6	5	0.83	0.094	0.81	Excellent
6	4 **	0.67	0.234	0.57	Fair

I-CVI—Item-level Content Validity Index. *P_c_* = [N!/A! (N − A)!] × 0.5^N^, probability of chance occurrence, where N = number of experts and A = number agreeing on relevance. *k** = (I-CVI − *P_c_*)/(1 − *P_c_*) kappa designating agreement on relevance; E_K_, evaluation criteria for kappa, described guideline by Cicchetti and Sparrow (1981). Fair = *K* of 0.40–0.59. Good = *K* of 0.60–0.74. Excellent = *K* of > 0.74. ** binomial variable (Polit et al., 2007).

## Data Availability

Data supporting reported results (Appendix A) can be found at: https://zenodo.org/record/7402378#.ZAdcUHZKjIW.

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
