# Peer review of "Development and Validation of Nutrition and Food Safety Educational Material for Fish Processors in Nigeria"

_ijerph, 2023, doi:10.3390/ijerph20064891_

Round 1

Reviewer 1 Report

This work developed and validated nutrition and food safety flipbook for the interactive training of small-scale fish processors in Nigeria. The manuscript was well written. However, there are some issues that should be addressed before it can be accepted. Is this kind of easy to understand teaching material only not available in Nigeria? Is it available in other parts of the world? If it is available in other regions, will others use the materials you developed? The contribution of your materials to foreign countries is beyond discussion. It is your advantage to develop easy to understand educational materials in combination with local characteristics, so, what localization work have you done for the educational materials you developed in Nigeria?

Author Response

Thank you for reviewing our manuscript titled Development and Validation of Nutrition and Food Safety Educational Material for Fish Processors in Nigeria. 

The reviewer's comments and recommendations geared towards improving the quality and publishability of this manuscript have been carefully considered and explained point by point in the response to the reviewer’s comments.

Reviewer 2 Report

1. Please write the main objective of your work in the abstract section.

2. In the introduction section authors should mention the contribution and motivation in the points.

3. why did the authors select the given problem, more detail in the result section should be required.

4. In the discussion section authors should compare their work to the other existing work for superiority.

5. In the conclusion section authors should mention the limitation of the work and also mention some new approaches in the future.

6.  in the references section authors should add some latest 2022 references.

7. please improve the English; language, there are a lot of grammatical errors.

Author Response

(The authors gave the same response as above.)

Round 2

Reviewer 1 Report

It can be accepted in the present form.

Reviewer 2 Report

Accepted